# Comparison of Acute Withdrawal and Slow Taper of Antiseizure Medications during Video Electroencephalographic Monitoring: Efficacy for Shortening of Hospital Stay

**DOI:** 10.3390/jcm10245972

**Published:** 2021-12-20

**Authors:** Ayako Motoki, Naoki Akamatsu, Tomoyuki Fumuro, Ayako Miyoshi, Hideaki Tanaka, Koichi Hagiwara, Shinji Ohara, Takashi Kamada, Hiroshi Shigeto, Hiroyuki Murai

**Affiliations:** 1Department of Clinical Medical Sciences, International University of Health and Welfare Graduate School of Medicine, Tokyo 107-8402, Japan; 18m3021@g.iuhw.ac.jp (A.M.); murai@iuhw.ac.jp (H.M.); 2Epilepsy Center, Fukuoka Sanno Hospital, Fukuoka 814-0001, Japan; aya.m@live.jp (A.M.); hideaki08seven@gmail.com (H.T.); hagiwarakyu@gmail.com (K.H.); shinji.ohara@gmail.com (S.O.); kamada_takashi@kouhoukai.or.jp (T.K.); shigeto217@gmail.com (H.S.); 3Department of Neurology, International University of Health and Welfare School of Medicine, Narita 286-8686, Japan; 4Department of Laboratory Medicine, International University of Health and Welfare School of Medical Sciences at Okawa, Okawa 324-8501, Japan; fumuro@iuhw.ac.jp; 5Division of Medical Technology, Kyushu University, Fukuoka 819-0395, Japan

**Keywords:** epilepsy, video-EEG monitoring, drug withdrawal, dose tapering, COVID-19 pandemic

## Abstract

Antiepileptic medications (ASMs) are withdrawn at the epilepsy monitoring unit to facilitate seizure recordings. The effect of rapid tapering of ASMs on the length of hospital stay has not been well documented. We compared the mean length of hospital stay between patients who underwent acute ASM withdrawal and slow dose tapering during long-term video electroencephalography (EEG) monitoring. We retrospectively investigated 57 consecutive patients admitted to the epilepsy monitoring unit regarding the mean length of hospital stay in the acute ASM withdrawal group (*n* = 30) and slow-taper group (*n* = 27). In the acute-withdrawal group, all ASMs were discontinued once the patients were admitted. In the slow-taper group, the doses of ASMs were gradually reduced by 15–30% daily. We also evaluated the safety of the acute-withdrawal and slow-taper protocols. The mean lengths of hospital stay were 3.8 ± 1.92 and 5.2 ± 0.69 days in the acute-withdrawal and slow-taper groups, respectively (*p* < 0.005). No severe adverse events, including status epilepticus, were observed. Acute ASM withdrawal has the advantage of significantly reducing the length of hospital stay over slow tapering, without any severe adverse effects.

## 1. Introduction

Long-term video-electroencephalography (EEG) is widely applied as a useful tool to identify epileptic foci in the preoperative evaluation of intractable epilepsy and to confirm the diagnosis of episodic symptoms, including psychogenic non-epileptic seizures [1,2,3]. The early capture of seizures during monitoring is important for reducing the burden on patients and hospitalization costs. Shorter periods of hospitalization are required, as we are currently in the middle of the coronavirus disease (COVID-19) pandemic [4].

Adverse events, such as status epilepticus, should be avoided while ensuring the diagnostic accuracy and efficacy of video-EEG monitoring [5,6,7]. Some studies have reported an association between rapid antiepileptic medication (ASM) dose reduction and the occurrence of adverse events, while others have found no association between rapid ASM dose reduction and the occurrence of adverse events [8,9,10].

Only a few studies have compared the duration of hospital stay between groups subjected to acute withdrawal and slow dose tapering [4,11]. In this study, we evaluated whether acute ASM withdrawal has the advantage of reducing the duration of hospital stay over slow tapering, and further evaluated the issue of safety.

## 2. Materials and Methods

### 2.1. Participants

Among the 78 consecutive patients admitted for EEG monitoring at Fukuoka Sanno Hospital, Fukuoka, Japan, between October 2017 and January 2020, excluding patients diagnosed with non-epileptic seizures (*n* = 12) and those with no change in ASM during monitoring (*n* = 9), 57 patients were retrospectively investigated regarding the length of hospital stay and latency of the first seizure after the start of monitoring in the acute-ASM-withdrawal (*n* = 27) and slow-taper (*n* = 30) groups. Data regarding patient demographics and ASM doses were collected from the medical records. The patients who underwent intracranial monitoring were excluded from hospital stay analysis (*n* = 11). The study protocol was approved by our Institutional Research Ethics Committee (approval no. 18-Ifh-015).

### 2.2. Method of Long-Term Video-EEG Monitoring

We performed long-term video-EEG monitoring using a scalp International 10–20 electrode placement system with additional bilateral anterior temporal electrodes. Invasive monitoring was performed with stereo-EEG or subdural grid electrodes. The patients were instructed to stay in bed during monitoring. Intravenous access was secured among all patients, and intravenous diazepam, fosphenytoin, and levetiracetam were administered when needed. One family member attended to the patient throughout the monitoring period to report the occurrence of seizures. The hospital staff included four epileptologists and five neurophysiology technicians who rotated during the daytime. During the nighttime and weekends, two nurses provided 24 h coverage, and an epileptologist was on call. Monitoring was continued until the epileptologist adjudged that sufficient diagnostic information had been obtained. The patients were discharged 24 h after ASMs had been restarted.

### 2.3. Antiseizure Medication Withdrawal/Taper Protocol

Acute withdrawal was defined as the total discontinuation of ASMs from the beginning of monitoring (Table 1). Slow-taper was defined as a gradual ASM dose reduction of 15–50% daily. The method of discontinuation and dose reduction was determined by the physician in charge. Patients admitted by an attending physician (N.A.) were allocated to the acute-withdrawal protocol, and other patients underwent slow-taper protocol. Patients with a history of status epilepticus were monitored without ASM dose reduction; thus, they were excluded from this study.

### 2.4. Statistical Analysis 

We analyzed the difference in latency of the first seizure and the average length of hospitalization between the two groups. The chi-squared test was used to compare categorical data between the two groups (sex; seizure type; history of febrile seizures; etiology of epilepsy; imaging findings; epilepsy syndrome; the number of antiepileptic drugs; and seizure clusters within 24, 48, and 72 h). An independent sample *t*-test was used to compare quantitative data between the two groups (age, age at onset of epilepsy, duration of epilepsy, seizure frequency, antiepileptic drugs being taken at admission, duration of long-term video-EEG monitoring, and the time to first seizure). All values are expressed as the mean ± standard deviation. The statistical analyses were performed using SPSS statistics version 27 (IBM Corporation, Armonk, NY, USA). In all cases, analysis items with a *p*-value < 0.05 were considered statistically significant.

## 3. Results

### 3.1. Baseline Demographic Profile

No statistically significant differences were observed between both groups in the baseline demographics of sex, duration of epilepsy, seizure frequency, history of febrile seizures, magnetic resonance imaging findings, or epilepsy syndrome (Table 2). However, there were significant differences in the duration and etiology of epilepsy. The distributions of daily ASM use were similar; however, only the levetiracetam dose was significantly higher in the slow taper group.

### 3.2. Comparison of Length of Hospitalization 

The mean length of hospital stay was significantly shorter in the acute-withdrawal group than in the slow-taper group (3.8 ± 1.92 vs. 5.2 ± 0.69 days, Table 3). The mean latency of first seizure onset was 33 ± 18 h (95% confidence interval (CI): 25–40 h, median: 27 h) in the acute-withdrawal group and 43 ± 34 h (95% CI: 30–55 h, median: 30 h) in the slow-taper group, which was shorter in the withdrawal group, but without statistical significance.

There were no correlations among time and the first seizure and monthly seizure frequency. Seizures occurred within 72 h in 83% of patients in the slow-taper group and 96% of patients in the withdrawal group, indicating that most seizures were captured within 72 h of admission. The frequency of seizures in the immediate pretest period did not correlate with the latency of seizure onset.

The number of the recorded seizure was similar in the acute-withdrawal group and in the slow-taper group (6.0 ± 6.9 vs. 5.2 ± 7.1). The number of generalized tonic–clonic seizure was 0.5 ± 1.0 (median: 0) in the acute-withdrawal group and 1.2 ± 1.9 (median: 1) in the slow-taper group, which was fewer in the withdrawal group, but without statistical significance. There were no significant side effects and/or drug intolerance after the re-introduction of the ASM.

## 4. Discussion

In the present study, the duration of hospitalization was shorter in the acute-withdrawal group than in the slow-taper group. Shortening of hospitalization has several advantages. In the era of the COVID-19 pandemic, there is limited availability of hospital beds for the care of patients with epilepsy, hence the need to shorten the duration of hospital stay as much as possible [4]. From an economic perspective, shortening video monitoring by one day would result in a cost cut of approximately 80,000 Japanese yen (USD 720) at our hospital. As the availability of video-EEG monitoring is limited in Japan, there is a long waiting list for monitoring. Shorter hospital stays for epilepsy monitoring would help us to clear the backlog and decrease the inconvenience imposed on patients and their caregivers [11]. The findings of our study are in accordance with those of a recent randomized controlled study that reported that rapid withdrawal had the advantage of shortening the monitoring duration [10].

For patients treated with ASMs with longer half-lives, such as phenobarbital and/or zonisamide, it is recommended that ASMs be discontinued before initiating video-EEG to capture seizures within a limited monitoring time [12]. In our study, only one patient was on phenobarbital, and seizures occurred within 3 days, and four patients on perampanel experienced seizures within 1 to 2 days. Further studies are necessary to investigate the prior ASM withdrawal protocol before hospital admission.

The most important concern in video-EEG monitoring is patient safety [2,5]. Previous studies have reported that seizure clusters increase in rapid ASM tapering protocols [3,6]. This raises concerns about the possibility of status epilepticus [5]. Recent studies have reported that rapid ASM withdrawal is safe in terms of the risk of status epilepticus [9,10,11]. In our study, there was no increase in number of generalized tonic clonic seizures in the rapid-withdrawal group. As the GTC is a risk factor of the sudden unexpected death in epilepsy [13], minimizing the occurrence of GTC is warranted during video EEG monitoring.

Another concern is that ASM withdrawal triggers non-habitual seizures and misleads the determination of seizure foci. Tapering of antiepileptic drugs increases the seizure frequency and secondary generalization but does not affect the pattern of seizure initiation or propagation [8]. Acute withdrawal of ASMs facilitated the diagnosis of the seizure onset zone in a recent study [4].

Our study had some limitations. The number of study patients was small. The allocation of patients to each withdrawal arm was not randomized. The patient’s demographics were not comparable in some parameters including numbers of ASMs, which may be related to the severity of epilepsy. The difference may have influenced the latency of seizure occurrence and monitoring length in our study. Further large-scale studies are required to confirm the findings of the present study.

## 5. Conclusions

The acute ASM withdrawal protocol may be used to shorten hospitalization for video-EEG monitoring without adversely affecting patient safety.

## Figures and Tables

**Table 1 jcm-10-05972-t001:** Antiseizure medication withdrawal/taper protocol.

Acute Withdrawal	Slow Taper
All the ASMs are stopped from Day 1	Reduction of each ASM with 15–50% daily, from Day 1 to the last day, until enough seizures are recorded

ASM; antiseizure medication.

**Table 2 jcm-10-05972-t002:** Demographic and clinical characteristics of patients (*n* = 57).

Demographic Characteristics	Total *n* = 57	Group A: Withdrawal *n* = 27	Group S: Slow Taper *n* = 30	*p*-Value
Sex				0.91
Male	27	13	14	
Female	30	14	16	
Age (years)				
Range	16–67	19–67	16–58	
Mean ±SD	33.5 ± 13.3	37.2 ± 14.5	30.2 ± 11.3	
Median	33.0	35.0	28.0	
Age at onset of epilepsy (years), mean ± SD	16.8 ± 11.9	22.0 ± 14.3	12.3 ± 7.0	0.004
Duration of epilepsy (years), mean ± SD	16.5 ± 12.5	14.7 ± 12.4	18.4 ± 12.4	0.26
Seizure type				0.01
FIAS	43	25	18	
GTCS	10	2	8	
Others	4	0	4	
Seizure frequency (months)				0.39
Range	0.08–100	0.17–60	0.08–100	
Mean ± SD	11.8 ± 20.3	9.3 ± 14.2	14.0 ± 24.6	
Median	4.0	4.0	4.0	
History of febrile seizures				0.85
Present	9	5	4	
Absent	44	20	24	
Unknown	4	2	2	
Epilepsy etiology				0.01
Hippocampal sclerosis/atrophy	3	2	1	
Encephalitis	7	0	6	
Brain tumor	0	0	2	
Others	11	3	9	
Unknown	34	20	14	
MRI findings				0.81
Negative	36	18	18	
Regional abnormality	18	8	10	
Unknown	3	1	2	
Epilepsy syndrome:				0.19
FLE	15	5	10	
TLE	34	20	14	
Generalized epilepsy	6	1	4	
Others	2	1	2	
Number of ASM(s)				0.03
1	6	6	0	
2	15	8	7	
3	23	11	12	
4	7	2	5	
5	4	0	4	
6	2	0	2	
Mean percentage reduction in antiepileptic drug doses from the baseline (%)	-	100 ± 0	76.6 ± 25.9(95% CI: 13.7–33.1)	0.00
Antiepileptic drugs at admission, mean dose ± SD (number of patients)				
	Phenytoin	281.3 ± 44.2 (2)	195.0 ± 42.0 (4)	0.07
	Sodium valproate	760.0 ± 296.7 (5)	700.0 ± 258.2 (4)	0.75
	Carbamazepine	566.7 ± 210.3 (12)	487.5 ± 196.2 (16)	0.32
	Gabapentin	2400 (1)	(0)	-
	Levetiracetam	1328.1 ± 489.2 (16)	2079.5 ± 835.9 (22)	0.001
	Lamotrigine	254.2 ± 110.0(6)	208.3 ± 106.8 (6)	0.48
	Clobazam	18.8 ± 10.3 (4)	23.9 ± 10.5 (9)	0.43
	Phenobarbital	60 (1)	(0)	-
	Zonisamide	(0)	265.0 ± 79.0 (4)	-
	Lacosamide	233.3 ± 111.8 (9)	283.3 ± 74.8 (15)	0.20
	Topiramate	300 (1)	325.0 ± 125.8 (4)	0.87
	Clonazepam	1.5 (2)	1 ± 0.71 (2)	0.50
	Perampanel	6.0 ± 2.8 (4)	4.7 ± 2.19 (15)	0.32

Abbreviations: SD, standard deviation; FIAS, focal impaired awareness seizure; GTCS, generalized tonic–clonic seizure; MRI, magnetic resonance imaging; FLE, frontal lobe epilepsy; TLE, temporal lobe epilepsy; ASM, antiseizure medication; CI, confidence interval.

**Table 3 jcm-10-05972-t003:** Length of hospital stay, time to first seizure, number of recorded seizures, and complications.

	Acute Withdrawal *n* = 27	Slow Taper *n* = 30	*p*-Value	Mean Difference (95% CI)
Hospital stay (days), mean ± SD	3.8 ± 0.70 (*n* = 24)	5.2 ± 1.97 (*n* = 22)	0.005	1.35 (0.44–2.3)
Time to first seizure (hours), mean ± SD; median	32.7 ± 18.7; 27.6	42.9 ± 33.8; 30.2	0.17	10.13 (−4.25 to 24.53)
First seizure in 4 h	0	0	-	-
First seizure in 24 h	11 (41%) (*n* = 27)	10 (33%) (*n* = 30)	0.56	-
First seizure in 48 h	22 (82%) (*n* = 27)	21 (68%) (*n* = 30)	0.32	-
First seizure in 72 h	26 (96%) (*n* = 27)	25 (83%) (*n* = 27)	0.12	-
No. of recorded seizures, mean ± SD; median	6.0 ± 6.9 (*n* = 27), 3	5.2 ± 7.1 (*n* = 30), 4	0.43	-
No. of recorded GTCs, mean ± SD; median	0.5 ± 1.0 (*n* = 27), 0	1.2 ± 1.9 (*n* = 30), 1	0.09	-
Status epilepticus	0	0	-	-
Other LTM-associated complications	0	0	-	-

Abbreviations: SD, standard deviation; CI, confidence interval; LTM, long-term electroencephalographic monitoring; GTC, generalized tonic clonic seizure.

## Data Availability

Data supporting reported results may be obtained upon request to the corresponding author.

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
