# Peer review of "Comparison of Acute Withdrawal and Slow Taper of Antiseizure Medications during Video Electroencephalographic Monitoring: Efficacy for Shortening of Hospital Stay"

_jcm, 2021, doi:10.3390/jcm10245972_

Round 1
Reviewer 1 Report
Dear author, the study is very interesting and useful for clinical application.
It would be useful to know if the patients had seizures control problems after re-introduction of the therapy or if they had side effects and / or drug intolerance.
I would be more specific on how many patients started the drug's reduction before the monitorig.
Author Response
We wish to express our appreciation to the reviewer for the insightful comments, which have helped us significantly improve the paper.
Point1:Dear author, the study is very interesting and useful for clinical application. It would be useful to know if the patients had seizures control problems after re-introduction of the therapy or if they had side effects and / or drug intolerance.
Response 1: Thank you for your comment. In accordance with the reviewer comment, we added the following sentence:
((at the “3 Results “section, third paragraph) There were no significant side effects and/or drug intolerance after the re-introduction of the ASM.
Point 2: I would be more specific on how many patients started the drug's reduction before the monitoring.
Response 2: Thank you for the comment. Of the 78 consecutive monitored patients during the study period, nine patients did not undergo ASM reduction, which is briefly stated in the “Materials and Method, Participants”.
We wish to thank the reviewer again for your valuable comment. We trust that the revised manuscript is suitable for publication.
Reviewer 2 Report
The authors present a retrospective study that analysed the length of hospital stay in patients with epilepsy admitted for video EEG monitoring. They found that patients in whom all antiepileptic drugs were discontinued on admission had a shorter hospital stay (3.8 days) than those in whom the antiepileptic drugs were gradually tapered (5.2 days), without occurence of status epilepticus in either group.
The question of this study has been investigated before, including in a prospective study (reference 11, the abstract in pubmed of this study does not state that it was a randomized controlled study as stated in the discussion).
The main limitation is that the decision to taper was not randomized or controlled, and that the two groups were not balanced in some aspects (more patients with "symptomatic" etiology and more patients with GTCS in the slow taper group, no patients on monotherapy in the slow taper group). The explorative statistical approach does not compensate for these limitations. The study therefore does not add much.
Author Response
Response to Reviewer 2 Comments
Point 1: The question of this study has been investigated before, including in a prospective study (reference 11, the abstract in pubmed of this study does not state that it was a randomized controlled study as stated in the discussion).
Response 1: Thank you for pointing out the miss-citation. The prospective randomized study was by Kumar et.al., and the reference number should be [10]. We corrected the citation accordingly.
Point 2: The main limitation is that the decision to taper was not randomized or controlled and that the two groups were not balanced in some aspects (more patients with "symptomatic" etiology and more patients with GTCS in the slow taper group, no patients on monotherapy in the slow taper group). The explorative statistical approach does not compensate for these limitations. The study therefore, does not add much.
Response 2: We appreciate the reviewer’s comment. The limitation of our report is mentioned in the last paragraph of the discussion. The randomized study is desirable from the evidence-based medicine point of view; however, it is more time and budget-consuming compared to a retrospective study.
 As the reviewer pointed out, there was some difference in demographic characteristics in each group, however, the overall severity of the epilepsy is comparable, including the duration of epilepsy, MRI findings, epilepsy classification, and dose of ASM.
 We believe that our retrospective observation has some value to add to the clinician, especially those treating patients at the epilepsy monitoring units.
We wish to thank the reviewer again for your valuable comment. We trust that the revised manuscript is suitable for publication.
Reviewer 3 Report
The manuscript submitted form Ayako Motoki et al., entitled "Comparison of acute withdrawal and slow taper of antiseizure medications during video- electroencephalographic monitoring: efficacy for shortening of hospital stay", is a short article. Interesting and original, but to improve the quality and the message I would suggest the authors to improve some critical points:
1) The authors should improve the introduction of the manuscript.
2) To improve the quality and the interest of the manuscript I would suggest the authors to insert figures that in summary explain the protocol that they established.
3) The authors in the discussion should mention also the limit (if there are) of this manuscript.
Author Response
Response to Reviewer 3 Comments
We wish to express our appreciation to the reviewer for the insightful comments, which have helped us significantly improve the paper.
Point 1: The authors should improve the introduction of the manuscript.
Response 1:  We carefully reviewed the introduction as suggested by the reviewers. The explanation of the background for this study may be too concise, we believe that the manuscript includes all the important points. We referred to the current utilization of video-EEG monitoring, safety issues, and length of monitoring. As many as 11 papers are listed as citations.
Point 2:To improve the quality and the interest of the manuscript I would suggest the authors insert figures that in summary explain the protocol that they established.
Response2 By the reviewer’s suggestion, we inserted the following table in the method.
|
Acute withdrawal |
Slow taper |
|
All the ASM are stopped from Day 1. |
Reduction of each ASM with 15-50% daily, from Day 1 to the last day, until enough seizures are recorded. |
Point 3: The authors in the discussion should mention also the limit (if there are) of this manuscript.
Response3: Thank you for your comment. The limitation of our study is the small number of studied patients and retrospective methodology rather than randomized study. We discussed the limitation in the last paragraph of the discussion.
We wish to thank the reviewer again for your valuable comment. We trust that the revised manuscript is suitable for publication.
Round 2
Reviewer 2 Report
The authors have corrected minor points, however the fundamental problem with this study is that the two groups are not comparable in key parameters, in particular in the number of AED (fast taper group mean of 2.34, slow taper group mean of 3.4, indicating that the slow taper group had more "severe" epilepsy).
Author Response
Response to Reviewer 2 Comments
Point: The authors have corrected minor points, however the fundamental problem with this study is that the two groups are not comparable in key parameters, in particular in the number of AED (fast taper group mean of 2.34, slow taper group mean of 3.4, indicating that the slow taper group had more "severe" epilepsy).
Response: We appreciate the reviewer’s comments pointing out the important limitation of our study. In accordance with the reviewer’s comments, we added the following sentence in the discussion (page6, line 175):
The patient’s demographics were not comparable in some parameters including numbers of ASMs, which may be related to the severity of epilepsy. The difference may have influenced the latency of seizure occurrence and monitoring length in our study.
We wish to thank the reviewer again for your valuable comment. We trust that the revised manuscript is suitable for publication.
Reviewer 3 Report
The authors satisfied all my concerns.
Author Response
Response to Reviewer 3 Comments
Point 1: The authors satisfied all my concerns.
Response 1: We wish to thank the reviewer again. We trust that the revised manuscript is suitable for publication.